# Metabolic and Antioxidant Responses of Different Control Methods to the Interaction of *Sorghum sudangrass hybrids*-*Colletotrichum boninense*

**DOI:** 10.3390/ijms25179505

**Published:** 2024-08-31

**Authors:** Jingxuan Xu, Junying Li, Hongji Wang, Xinhao Liu, Zhen Gao, Jie Chen, Yuzhu Han

**Affiliations:** 1College of Animal Science and Technology, Southwest University, Chongqing 402460, China; xujingxuan0128@163.com (J.X.); lijunying0312@163.com (J.L.); whongji123@163.com (H.W.); 15963138985@163.com (X.L.); xndxgz@163.com (Z.G.); 18326204129@163.com (J.C.); 2Chongqing Key Laboratory of Herbivore Science, Chongqing 402460, China

**Keywords:** *Colletotrichum boninense*, biological control, *Trichoderma harzianum*, metabolism analysis

## Abstract

*Colletotrichum boninense* is the main pathogenic fungus causing leaf spot disease in *Sorghum sudangrass hybrids*, which seriously impairs its quality and yield. In order to find an efficient and green means of control, this study used the agar disk diffusion method to screen for a fungicide with the strongest inhibitory effect on *C. boninense* from among several bacteria, fungi, and chemicals. Then, the changes in the plant’s antioxidant system and metabolic levels after treatment were used to compare the three means of control. The lowest inhibitory concentration of Zalfexam was 10 mg/mL, at which point *C. boninense* did not grow, and the inhibition rates of *Bacillus velezensis* (X7) and *Trichoderma harzianum* were 33.87–51.85% and 77.86–80.56%, respectively. Superoxide dismutase (SOD) and chitinase were up-regulated 2.43 and 1.24 folds in the *Trichoderma harzianum* group (M group) and SOD activity was up-regulated 2.2 folds in the *Bacillus velezensis* group (X7 group) compared to the control group (CK group). SOD, peroxidase (POD), and chitinase activities were elevated in the Zalfexam group (HX group). The differential metabolites in different treatment groups were mainly enriched in amino acid metabolism and production, flavonoid production, and lipid metabolism pathways. Compared with the diseased plants (ZB group), the M, X7, HX, and CK groups were co-enriched in the tryptophan metabolic pathway and glutamate–arginine metabolic pathway, and only the CK group showed a down-regulation of the metabolites in the two common pathways, while the metabolites of the common pathways were up-regulated in the M, X7, and HX groups. In addition, the salicylic acid–jasmonic acid pathway and ascorbic acid–glutathione, which were unique to the M group, played an important role in helping *Sorghum sudangrass hybrids* to acquire systemic resistance against stress. This study fills the gap in the control of *Colletotrichum boninene*, which causes leaf spot disease in *Sorghum sudangrass hybrids*. This paper represents the first reported case of biological control for leaf spot disease in *Sorghum sudangrass hybrids* and provides a reference for the control of leaf spot disease in *Sorghum sudangrass hybrids* as well as other crops infected with *Colletotrichum boninense*.

## 1. Introduction

The *Sorghum sudangrass hybrid* (*Sorghum bicolor* × *Sorghum sudanense*) is a high-quality forage grass mainly grown in Southwest China, and is an annual grass crop that is a cross between *Sorghum bicolor* and *Sorghum sudanense* [1]. The *Sorghum sudangrass hybrid* has excellent characteristics such as high yield, good quality, and strong adaptability, which are highly valued by the livestock industry at home and abroad. However, with the rapid development of the livestock industry, the demand for higher yielding and high quality *Sorghum sudangrass hybrids* is also increasing [2]. In recent years, leaf spot disease in *Sorghum sudangrass hybrids* grown in Bazhong City, Sichuan Province, China, has been as high as 85–88% [3], seriously affecting forage quality and yield, causing economic losses. The laboratory pre-studied leaf spot in *Sorghum sudangrass hybrids* grown in the Bazhong area and found that the main pathogens were *Colletotrichum boninense*, *Nigrospora sphaerica*, and *Didymella corylicola*, among which *C. boninense* had the strongest pathogenetic abilities [3]. *Colletotrichum boninense* is a species within the genus *Colletotrichum*, specifically belonging to the subgroup *Glomerella* in the subclass *Hypocreomycetidae* of the family *Ascomycota* (Incertae sedis), under the family *Glomerellaceae.* This widely distributed fungus has a broad range of hosts and is known to cause anthracnose in various crops, leading to symptoms such as leaf spot, leaf blight, flower rot, fruit rot, and branch blight. These manifestations negatively impact plant growth and development, ultimately reducing crop quality [4]. *C. boninense* has been reported to infect a variety of plants such as rose [5], coffee tree [6], and goldthread [7], causing severe economic losses.

Current control methods for diseases in *Sorghum sudangrass hybrids* are mainly focused on the use of chemicals. In China, rust has been reported in areas with high humidity. Chemical control methods have included spray treatments with streptomycin in combination with guanosine and aminoglycosides, as well as spray treatments with Hogoli WP in combination with 2% powdered rust emulsion [8]. Chemical control methods offer the benefits of cost-effectiveness and rapid efficacy, enabling swift management of pathogenic bacteria infestation and damage within a short timeframe. This ultimately helps minimize losses associated with the disease [9]. For example, Barley seed treated with 300 mg/L mancozeb prevented spot blotch and increased chlorophyll production in the plant [10]. A 20% pyraclostrobin–tebuconazole suspension provided good control of chrysanthemums powdery mildew in field-grown plants [11]. Meanwhile, chemical fungicides are an extremely effective tool (at least in the short term) for reducing disease incidence in crops, however, they have harmful effects on beneficial plant microbiota, the health of humans and other animals, and on the environment [12]. With the rapid advancement of modern biotechnology, biological control agents have emerged as a significant method for managing plant diseases. These techniques are environmentally friendly, non-toxic, and sustainable, leading to their widespread investigation and application in the realm of plant disease management [13]. However, biocides are often expensive compared to synthetic biocides [14]. *Trichoderma* is a quintessential alternative to chemicals. Its mycoparasitic abilities, along with antibiosis and induction of host immunity, are the main mechanisms by which this fungus performs biocontrol and it has been identified as a potential biocontrol agent [15]. *Trichoderma pseudokoningii* improves control of cucumber wilt by increasing the capacity of the antioxidant system of cucumber seedlings and promotes seedling morphogenesis [16]. *T. harzianum* promotes the growth of red kidney bean and inhibits root rot by improving plant antioxidant enzyme activity and regulating the rhizosphere microbial community [17]. Studies on Atractylodis root rot have also suggested that *Bacillus spp.* and *T. harzianum* complex could be considered as potential future biocontrol agents against Fusarium root rot [18]. Inoculation with *B. velezensis* has also been reported to control Potato common scab in potatoes [19].

*C. boninense* is highly pathogenic, infects a wide range of plant hosts, and is the most damaging to *Sorghum sudangrass hybrids*. Therefore, based on the five fungal strains and four bacterial strains with antagonistic abilities screened by the research group in the previous stage, and four chemical agents commonly used for the treatment of leaf spot disease, the present study was conducted to screen out the most effective fungi or agents for each of the antagonistic tests on *C. boninense*. The metabolic levels of healthy, control, and diseased plants were then analysed and compared using untargeted metabolomics and liquid chromatography–tandem mass spectrometry (LC-MS/MS). We investigated the efficacy of different fungicides in controlling *C. boninense* leaf spot and revealed the differences in metabolites, pathways, and antioxidant enzymes between diseased and treated plants, which will provide a theoretical basis and reference for the control of *C. boninense* and other diseases caused by *C. boninense* in the field.

## 2. Results

### 2.1. Initial Screening of Chemical Fungicides

The results of different concentration gradient plate tests showed that all four chemicals, pyraclostrobin, zalfexam, thiophanate–methyi, and mancozeb, were effective against the pathogen *C. boninense* (Appendix A, Table A1). In a detailed analysis of the minimum inhibitory concentration and inhibition rates of various chemical fungicides, zalfexam exhibited a minimum inhibitory concentration of 10 mg/mL, with inhibition rates ranging from 77.35–82.65% at 5 mg/mL and 66.11–72.55% at 0.1 mg/mL (Figure 1b and Figure 2d). The growth of *C. boninense* on Thiophanate–methyl medium at concentrations of 10 mg/mL, 5 mg/mL, and 0.1 mg/mL showed no significant difference during the first three days (Figure 1a and Figure 2f). The inhibition of the pathogens after seven days of incubation ranged from 79.36% to 83.60%, 83.46% to 85.06%, and 84.85% to 86.45% at these concentrations. However, at a concentration as high as 20 mg/mL, there was no growth observed (Figure 1b and Figure 2f). Therefore, the lethal concentration of thiophanate–methyl against the pathogens was higher than that of the zalfexam combination. After 7 days of being cultured on a medium containing pyraclostrobin at concentrations of 20 mg/mL, 10 mg/mL, 5 mg/mL, and 0.1 mg/mL, significant differences in colony diameters of pathogenic fungus were observed (*p* < 0.05). The mean inhibition rates were 100%, 75.15–79.15%, 71.35–72.73%, and 54.53–57.55%, respectively (Figure 1b and Figure 2c). The mean inhibition rates of *C. boninense* were 100%, 58.01–63.35%, and 24.19–41.66%, respectively, after 7 days of incubation in the media with the addition of 10 mg/mL, 5 mg/mL, and 0.1 mg/mL of mancozeb (Figure 1b and Figure 2e). Except for the lowest inhibitory concentration, which was consistent with zalfexam, the low concentration of mancozeb had the worst inhibitory effect on the pathogen among the four agents, and the combination of the criteria of inhibition rate and lowest lethal concentration led to the preliminary selection of zalfexam as the chemical fungicide.

### 2.2. Screening Results of Bacterial Biocides

The results from the plate line confrontation of bacteria (Appendix A, Table A1) indicated that X7, X15, X12, and X21 exhibited inhibition rates of 46.80–55.32%, 32.21–38.71%, 33.05–54.89%, and 27.26–37.98% at 3 days. On the third day, all four bacteria showed a significant inhibitory effect on the pathogen *C. boninense*. However, there were no significant differences in inhibitory effects among the bacteria. After 7 days of growth, there was no significant difference in the growth diameter of the pathogens when confronted with the four bacteria. Notably, X7 demonstrated the most significant inhibitory effect, performing significantly better than the other three bacteria, with inhibition rates ranging from 33.87–51.85% (Figure 1c and Figure 2b). Continuing the culture until the 15th day, the average diameters of the four bacteria (X7, X15, X12, X21) when confronted with the pathogen were observed to be 6.4 cm, 7.7 cm, 8.2 cm, and 8 cm. At this time, the diameter of the control colony was 9 cm, indicating that there was almost no inhibitory effect of X12 and X21 on the pathogenic fungi at 15 days. Additionally, strain X7 was able to achieve a bacterial inhibition rate of 26.944–31.576% (Appendix A, Figure A1). In conclusion, strain X7 was selected as the most effective bacterial fungicide against pathogenic *C. boninense*.

### 2.3. Screening Results of Fungicides

The results of the fungal confrontation experiment (Appendix A, Table A1) indicated that in the confrontation test involving strains Z7 and Z9, *C. boninense* exhibited dominant growth, while Z7 and Z9 showed minimal to no growth. Strain Z19 exhibited some growth on the third day, but did not hinder the growth of *C. boninense* (Figure 1d and Figure 2a). After 7 days of incubation, inhibition bands were observed on the medium inoculated with strain Z19 against pathogenic fungi. The inhibition rates ranged from 7.66% to 23.38%, but the growth of pathogenic fungi still prevailed. When exposed to *T. harzianum*, *C. boninense* exhibited limited growth and *T. harzianum* showed significant inhibition. The inhibition rates were 32.17–47.45% and 77.96–80.56% on day 3 and day 7, respectively (Figure 1d and Figure 2a). The growth trend of Z8 and the pathogen showed similarities at 3 days, with both having similar diameters, albeit with a slight advantage for Z8. By day 7, Z8 had taken the dominant position, exhibiting a clear inhibitory effect on the pathogen, resulting in inhibition rates of 13.72–39.98% at 3 days and 37.05–85.17% at 7 days (Figure 1d and Figure 2a). Therefore, *T. harzianum*, with a higher average inhibition rate, was selected as the candidate for fungal control agent.

### 2.4. Measurement of Biomass of Different Treatment Groups Treating Sorghum sudangrass hybrids Seedlings

In the pot test, *Sorghum sudangrass hybrids* from the ZB group displayed disease spots on leaves and rhizomes 5 days after inoculation, with an incidence rate of 67.8%. Subsequently, the incidence rates for the M group, HX group, and X7 group 7 days after inoculation were 12%, 22.8%, and 24% respectively. The disease symptoms in the different treatment groups were consistent, characterized by reddish-brown spots starting from the edges of the leaves. The spots exhibited varying shades of color across different parts of the affected area. The growth and development of *Sorghum sudangrass hybrids* seedlings varied among the four treatment groups, affecting them differently compared to the CK group (Table 1). Diseased plants showed a mean height of only 34.33 ± 9.018 cm, with lower biomasses in terms of root length, root number, volume, plant height, and leaf length than plants in other treatment groups. The average plant height, root number, root length, and root volume did not significantly differ among the CK, M, X7, and HX groups. However, the root length of plants in the HX group treated with zalfexam was higher than in other groups, and the root volume and leaf length of plants in the M, X7, and HX groups were higher than in the healthy group. These findings suggest that the three different treatments may have a growth-promoting effect on the plants.

### 2.5. Analysis of Enzyme Activities in Different Treatment Groups of Treated Sorghum sudangrass hybrids Seedlings

To investigate the impact of fungal group M, bacterial group X7, and chemical agent HX on leaf spot disease in *Sorghum sudangrass hybrids*, we analyzed the levels of SOD, POD, and catalase (CAT) in the leaves of *Sorghum sudangrass hybrids* one week after inoculation with the disease fungus solution across different treatment groups. Our results showed a significant difference in CAT enzyme activity between the HX group and the other four groups, with the lowest activity observed in the HX group. While the differences in CAT enzyme activity between the CK, X7, and M groups were not significant, they were still significantly different from the ZB group. In the variation of POD enzymes (Figure 3b), all four treatment groups showed significant differences compared to the ZB group and exhibited lower levels than the diseased plants. The ZB group had 8.28-fold higher POD enzymes than the CK group, suggesting that the pathogen *C. boninense* poses a substantial threat to the antioxidant system of *Sorghum sudangrass hybrids*. The content of POD activity in the HX group was 3-fold and 1.6-fold higher than that in the X7 and M groups. The results presented in Figure 3c indicate that SOD enzyme activity was lower in the HX group compared to the ZB group, with no significant difference observed with the CK group; however, SOD enzyme activity was significantly higher in the M and X7 groups. Given that SOD enzyme activity is a crucial indicator of plant stress response, this study suggests that biocontrol fungi may be more effective in treating leaf spot disease in *Sorghum sudangrass hybrids*. Although the HX group showed significant differences from the biocontrol fungi treatments (M and X7), it exhibited lower SOD enzyme activity. This implies that biocontrol fungi may stimulate the plant’s antioxidant system more effectively than chemicals, leading to reduced accumulation of reactive oxygen species (ROS) and enhanced plant resistance. Furthermore, plants treated with the chemical zalfexam displayed higher SOD enzyme activity compared to healthy plants, indicating that the chemical zalfexam is partly helpful in improving plant resistance. The activity of SOD enzyme increased significantly in the M and X7 groups, while POD enzyme activity showed a greater increase in the HX group. The pathogenic fungus *C. boninense* had a more pronounced impact on the SOD and POD enzyme levels of the plants. On the other hand, CAT enzyme content exhibited less variation among the groups, although notable differences were still observed. Plants employ various defense mechanisms to safeguard themselves against different pathogens. Chitinase and dextran are components found in fungal cell walls, allowing plants to directly target the structural elements of fungi [20]. Plant chitinases are activated in response to pathogen-induced stress, underscoring their crucial role in combating fungal infections [21]. In the chitinase assay (Figure 3d), the enzyme activity of the ZB group was 2.23, 1.74, 1.47, and 2.16 times higher than that of the X7, M, HX, and CK groups. The ZB group exhibited significant differences compared to all other groups, with the HX group showing the highest increase in enzyme activity across the three treatment groups, followed by the M group.

### 2.6. Metabolomic Analysis

To investigate metabolite differences among four different groups (CK, M, X7, and HX) and the ZB group, the metabolism of *sorghum sudangrass hybrids* plants treated with five different treatment groups (CK, M, X7, HX, ZB) was analyzed. Principal Component Analysis (PCA) is an unsupervised pattern recognition method used for statistically analyzing multidimensional data. It reveals the internal structure among multiple variables by extracting a few principal components that retain as much information as possible from the original variables while being uncorrelated with each other [22]. The 3D-PCA plot (Figure 4a) indicated that the first and second principal components accounted for 44.24% and 20.37% of the total variance, respectively, demonstrating significant differences in metabolic profiles among the sample groups. Orthogonal Partial Least Squares Discriminant Analysis (OPLS-DA), a multivariate statistical method with supervised pattern recognition, effectively screens for differential metabolites by eliminating irrelevant effects [23]. Model plots generated from OPLS-DA analysis of CK vs. ZB, X7 vs. ZB, M vs. ZB, and HX vs. ZB pairwise comparisons (Figure 4b) showed well-performing models, with high Q values close to 1. The significant separation observed in the OPLS-DA score plots among different comparison groups further supports the significant differences in metabolic profiles following various treatments. A total of 1456 different metabolites were detected through untargeted metabolomics analysis. The criteria for screening metabolic differences included a *p*-value (*p* < 0.05). Ultimately, 977 differential metabolites were identified, with 243 in the CK vs. ZB group, 432 in the X7 vs. ZB group, 177 in the M vs. ZB group, and 290 in the HX vs. ZB group. These differentially metabolised species were classified as Benzenoids, Lipids and lipid-like molecules, Organic acids and derivatives, Organoheterocyclic compounds, Phenylpropanoids and polyketides, Organic oxygen compounds, and seven other categories (Figure 5a).

In the KEGG enrichment analysis (Figure 5c), amino acid substances were found to be more abundant and serve as primary metabolites in the metabolic pathway. Plants are capable of synthesizing all these amino acids. In addition to their importance in plant growth and development, growing evidence underlines the central role played by amino acids and their derivatives in regulating several pathways involved in biotic and abiotic stress responses [24]. Amino acid metabolism in the CK vs. ZB group primarily focused on arginine and proline metabolism, cysteine and methionine metabolism, as well as histidine and tryptophan metabolism. On the other hand, in the HX vs. ZB group, the emphasis was on alanine, aspartate and glutamate metabolism, tryptophan metabolism, arginine and proline metabolism, and starch and sucrose metabolism. The M vs. ZB group showed a concentration on pathways related to alanine, aspartate and glutamate metabolism, arginine biosynthesis, as well as pantothenate and coenzyme A biosynthesis. Additionally, the X7 vs. ZB group demonstrated pathways involved in tryptophan and histidine metabolism, alanine, aspartate and glutamate metabolism, and unsaturated fatty acid biosynthesis. Some common pathways across the comparison groups included arginine metabolism, while pathways related to tryptophan, alanine, aspartate, and glutamate metabolism showed partial overlap among the four groups. In addition, the phenylalanine pathway and flavonoid biosynthesis are also crucial and play a variety of regulatory roles in plant growth and development, and have antioxidant and anti-inflammatory properties [25]. This study primarily focused on the production of naringenin, chlorogenic acid salt, and ferulic acid. The volcano plot was utilized to identify important metabolites, as illustrated in Figure 5b. Chloramphenicol was found to be the most downregulated in the ZB vs. CK group, while the most significantly altered substances were upregulated in the ZB vs. HX, ZB vs. M, and ZB vs. X7 groups. These are 2-Naphthalenesulfonic acid, Propentofylline, and n-Octyl-beta-D-thioglucopyranoside. In comparison to the ZB group, the CK group showed 109 up-regulated and 134 down-regulated metabolites; the X7 group had 294 up-regulated and 138 down-regulated metabolites; the M group had 95 up-regulated and 71 down-regulated metabolites; and the HX group had 174 up-regulated and 116 down-regulated metabolites. A clustering heat map (Figure 6) was used to analyze the accumulation pattern of metabolites in different treatment groups. Cluster analysis revealed that there were 56 metabolites common to all four groups (CK vs. ZB, X7 vs. ZB, M vs. ZB, and HX vs. ZB). Additionally, 32 unique metabolites were found in the HX group, 10 in the M group, and 18 in the X7 group.

To gain insights into the metabolism of *Sorghum sudangrass hybrids* following various treatments, pathway maps were constructed for four treatment groups (CK vs. ZB, X7 vs. ZB, M vs. ZB, and HX vs. ZB) using KEGG enriched pathways and metabolite analyses. Key pathways identified included amino acid synthesis and metabolism, flavonoid synthesis, tricarboxylic acid cycle, and lipid metabolism. The CK vs. ZB, HX vs. ZB, M vs. ZB, and X7 vs. ZB treatment groups focused on tryptophan metabolism and arginine–glutamate metabolism pathways. Notably, arginine and its derivatives were upregulated in the HX vs. ZB, M vs. ZB, and X7 vs. ZB groups, while arginine derivatives were downregulated in the CK vs. ZB group. In tyrosine metabolism to produce catecholamines, the pathway did not contain the HX vs. ZB group, the X7 vs. ZB group did not form a pathway for the production of the flavonoid naringenin, and the HX vs. ZB and M vs. ZB groups all had the formation of ferulic acid and Cilostazol, which were shown to be up-regulated. In addition, the M vs. ZB group had pathways for the production of arachidonic acid and jasmonic acid in lipid metabolism, both of which showed up-regulation, while the X7 vs. ZB group showed the production of arachidonic acid and the HX vs. ZB group showed the production of jasmonic acid. The study concluded that the different treatment groups all responded differently to stress, but the M vs. ZB group was involved in a richer regulation of metabolic pathways.

## 3. Discussion

### 3.1. Plant Fungal Disease Control Tools Are Diverse

*Colletotrichum boninense*, the major pathogen of *Sorghum sudangrass hybrids* leaf spot disease, caused severe yield losses in *Sorghum sudangrass hybrids* in Sichuan, China, and the prevalence of *Sorghum sudangrass hybrids* leaf spot disease in Bazhong City, Sichuan Province, was as high as 85–88%. *Colletotrichum boninense* was shown to significantly affect plant growth through back-joining experiments, with plant height reduced by 25% and root length, root number and volume, leaf size, and stem diameter all significantly lower than those of healthy plants [3]. This is similar to the results of our study, where the biomass of diseased plants was all significantly lower than that of plants in the other treatment groups. This also indicates that our treatment groups (HX, X7, M) were effective.

At present, chemical fungicides are still the more important and effective means of controlling fungal diseases in plants, and farmers rely heavily on chemicals to prevent food losses due to fungal diseases [26]. In a study of anthracnose on mung beans, a 25% propiconazole emulsion was identified as the most effective in suppressing the growth of anthracnose mycelium on mung beans [27]. All the four chemical fungicides selected in this study had a good fungicidal inhibitory effect on *C. boninense*. The lowest inhibitory concentration was 20 mg/mL for pyraclostrobin and thiophanate–methyl, while it was 10 mg/mL for zalfexam and mancozeb. Notably, mancozeb showed a weaker inhibitory effect at lower concentrations. However, chemical agents have high environmental pollution and residue rates, and biocide residues are less harmful to organisms and the environment, and are relatively safe to use even before harvest [14]. In terms of bacteriological control, *Bacillus velezensis* effectively halted the growth of *C. gloeosporioides*, inducing noticeable abnormalities such as hyphal breakage and distortion, thereby curtailing the pathogen’s virulence. A 50–100 times dilution of *B. velezensis* fermentation broth, applied every two to three days, served as an efficient protective layer for walnut leaves and fruits against *C. gloeosporioides* infection [28]. This study is consistent with the results of our bacterial fungicide screening. In addition, in the control of other plant diseases, *Bacillus velezensis JCK-7158* is being considered as a novel biocontrol agent for managing fusarium head blight (FHB) [29]. In a study, inoculation of potato with *Bacillus velezensis Y6* significantly reduced the incidence of potato scab [19]. Our research also demonstrated that *Bacillus velezensis-X7* exhibited promising inhibition of the pathogen, indicating its potential in preventing and controlling leaf spot disease in *Sorghum sudangrass hybrids*. In terms of fungal control, *Trichoderma harzianum* CGMCC20739 (Tha739) has shown potential in combating apple bitter rot caused by Colletotrichum gloeosporioides. Tha739 could control apple bitter rot and maintain the nutritional quality of the fruit. Thus, *T. harzianum* Tha739 is a potential biocontrol agent for harvested apples [30]. The biocontrol effect of *Trichoderma harzianum* has been documented in combating diseases caused by fusarium stalk rot [31] and botrytis cinerea [32]. In this study, *Trichoderma harzianum* exhibited a significant inhibitory effect, achieving up to 80.56% pathogen inhibition. This shows great potential for the biological control of *Trichoderma harzianum* and is the first example of a biological control agent being used on *Sorghum sudangrass hybrids*.

### 3.2. Antioxidant Enzymes Have an Important Role in Plant Response to Pathogen Stress

The antioxidant enzyme system in plants plays a crucial role in combating damage from membrane lipid peroxidation and oxidative stress caused by the accumulation of reactive oxygen radicals (ROS) in adverse conditions [33]. Antioxidant enzymes, including SOD, POD, and CAT, play a crucial role in plants by scavenging excess reactive oxygen species to mitigate environmental stress. This ability to withstand environmental stress is a key factor in evaluating plant disease resistance [34]. SOD is a major scavenger of free radicals as it converts the free radicals of superoxide anion into H_2_O_2_, which is then decomposed into harmless water by CAT and POD [35]. POD utilizes H_2_O_2_ to oxidize harmful substances like phenols, converting them into quinone compounds. On the other hand, CAT acts as a conjugating enzyme with iron–porphyrin as a cofactor, reacting with H_2_O_2_ to generate iron peroxide reactive species that facilitate H_2_O_2_ oxidation [36]. SOD enzyme activity showed a significant increase in the M and X7 groups, while in the HX group, it was POD activity and chitinase that saw a significant increase. The activities of SOD, POD, and chitinase showed greater variability in *C. boninense*-infected plants. Based on previous research [37], it was found that harpin led to higher antioxidant accumulation in young jujube leaves when compared to carbendazim. Carbendazim notably decreased antioxidant accumulation and total antioxidant capacity in young jujube leaves, particularly within the initial 7 days. This suggests a dual effect of the chemicals on antioxidant capacity, both decreasing and increasing it. Moreover, the application of zalfexam in this study resulted in the enhanced antioxidant capacity of the plants. *Trichoderma harzianum* has been reported to significantly increase H_2_O_2_ content and the activity of antioxidant enzymes such as catalase (CAT) and ascorbate peroxidase when used in conjunction with chemicals [38]. But, the rise in the activity of CAT enzyme was not significant in the treatment groups in the present study. One study reported that overexpression of Tachi in soybean caused increased reactive oxygen species (ROS) levels as well as peroxidase (POD) and catalase (SOD) activities, decreased malondialdehyde (MDA) content, along with a diminished electrolytic leakage rate after *S. sclerotiorum* inoculation [39]. Wang et al. [40] found that the antioxidant enzyme activities (SOD, POD, and CAT) in *S. asari* decreased significantly, while the MDA content and electrolytic leakage rate increased when treated with antagonist *Trichoderma harzianum* strains (A26 and B30). The study hypothesized that the alterations in POD and CAT enzyme activities could be linked to changes in MDA content and electrolytic leakage rate. It has been shown [41] that the enhancement in root growth and development as well as the increase in *Trichoderma harzianum* led to an increase in glutamate content in plant tissues and acted as an effective scavenger of free radicals, which is consistent with the results of the enhancement of the glutathione–ascorbic acid cycle in plants after treatment in group M in the present study. In a literature report [42] on the effect of inoculation with a strain of *Bacillus subtilis* on the antioxidant metabolism of grafted tomatoes, it was found that SOD and CAT enzyme activities were reduced in the first two weeks after grafting, because grafting can also induce a stress effect, and this is similar to the results of our study on SOD, where the X7 group was subjected to stress, manifesting as an increase in the enzyme activity of SOD. A study demonstrated that the application of *Bacillus* spp. augmented the antioxidant defense mechanisms in infected rice, consequently mitigating oxidative stress induced by Botrytis cinerea and impeding its progression [43]. These findings align with the elevated enzymatic activity of POD observed in the X7 group.

In summary, the HX group demonstrated the increased activity of POD and chitinase enzymes. The M group exhibited a more pronounced elevation in SOD and chitinase activities. The X7 group primarily showed enhanced activity in SOD. Notably, SOD serves as the principal free radical scavenger, with its activity enhancement being more important. Given that the M group displayed a significant uptick in SOD enzyme activity, it is posited that this group exerts a more substantial influence on antioxidant enzyme properties.

### 3.3. The Influence of Different Treatment Methods on the Important Metabolites of Sorghum sudangrass hybirds

In this study, we made heat maps of the common substances of the four groups and the unique substances of each group. Based on the heat maps, we found the following problems.

Compared with the ZB group, 23 substances, such as lidamidine, showed the same downward trend as CK after being treated with HX, M, and X7. Formononetin is a phytoestrogenic member of the flavonoid family, which has pharmacological effects such as anti-oxidation, anti-hypertension, anti-tumor, and anti-infection effects [44]. After infection, the expression of this substance was down-regulated, and the content of formononetin in each treatment after treatment was maintained at a high level, similar to that in the CK group, which means that the three treatments are effective in the treatment of *Sorghum sudangrass hybirds*. Emodin is a multi-effect molecule with diuretic, vasodilator, antibacterial, antiviral, anti-ulcer, anti-inflammatory and anti-cancer effects [45]. Chloramphenicol can inhibit the formation of the attachment membrane of blast fungi, thus inhibiting the infection of pathogenic fungus to rice [46] and the up-regulation of these substances is also related to the effective treatment of plants. Studies have shown that usnic acid and vulpinic acid have strong antibacterial activity against the subspecies of Bacillus michiganensis [47]. In our study, usnic acid was down-regulated, except in the ZB group. It is reasonable to guess that the reason is that our diseased grass has been effectively treated, and the plants do not need to secrete usnic acid for antibacterial purposes. Hui Bing et al. [48] studied tomato bacterial wilt and isolated an endophytic strain NEAU-CP5 with strong antagonistic activity from tomato seeds. NEAU-CP5 can produce known antimicrobial metabolites, including ring (leucylprolyl), ring (phenylalanyl-prolyl), etc.

In this study, both of these substances, along with lactic acid [49], which promotes plant growth, were down-regulated. The substances mentioned above all play an antibacterial and anti-infection role in the plant. In the HX, X7, and M groups, these substances are as down-regulated as the healthy *Sorghum sudangrass hybirds* in the CK group, which makes it obvious that our treatment is effective and can resist pathogen attacks. At the same time, we found that 15 substances were down-regulated in the X7 group and down-regulated in other groups. The ZB group was the most down-regulated, and HX, M groups, and CK groups showed similar down-regulated levels (Figure 6a). It has been shown [50] that endophytic Bacillus can enhance the biomass and bioactive metabolites of Garan, and the secondary metabolite content may contribute to the sustainable cultivation of economically important medicinal plants. *Bacillus pumilus* can alleviate drought stress and increase the accumulation of metabolites in licorice and *Bacillus microus* can improve the growth of *G. uralensis* under drought stress by changing the accumulation of antioxidants and increase the content of glycyrrhizin through the incremental expression of key enzymes [51]. The upregulation of these 15 substances in this study may be related to the special function of *Bacillus-X7*. Hesperidin has antioxidant, anti-inflammatory, and antibacterial activities [52]. The contents of methylparaben [53], which can enhance the overall adaptability of plants through disease resistance and allelopathy, were all increased. These substances have a positive effect on plant growth and disease resistance and can help X7 resist the attack of pathogens. The down-regulation of this substance in group ZB proves that pathogenic attacks reduce the content of this substance, and thus it becomes infected.

In addition, many unique functional compounds were detected in the X7 group in this study compared to the ZB group. Prunasin is a cyanogenic glycoside known for its role in the defense of herbivores and other plants [54]. In this study, this substance was up-regulated and it was speculated that X7 treatment enhanced the accumulation of this substance in the grass to resist pathogen attacks. Adenine substances directly act on the electron transport system complex to regulate the mitochondrial electron transport system and mitochondria-derived ROS production, thus directly affecting plant development [55]. The downregulation of this substance in this study may be related to the increased antioxidant activity in the X7 group. In summary, we hypothesized that X7 mainly defends *C. boninense* by accumulating these active substances compared with chemotherapy and *T. harzianum* treatment.

In the ZB and HX groups, ursolic acid is a triterpene with known antibacterial effects, which naturally exists in plants and has potential biological activities, such as anti-tumor, anti-viral, and anti-bacterial activities [56]. The upregulation of this substance indicated that the treatment of zolether–deisen could induce an increase in the production of this substance naturally occurring in plants, so as to resist the infection and destruction of pathogens. The expression of soyasapogenol A [57], which has anti-inflammatory, anti-mutagenesis, anti-cancer, anti-microbial, and liver and cardiovascular protective activities, is down-regulated. Therefore, we believe that imidazole–Sengen, which plays a disease-resistant role in the HX treatment group, can resist the attack of most pathogens by inducing the formation of antibacterial substances in plants. Thus, the variety and content of active substances secreted by the plant itself are reduced.

In the ZB and M groups, studies have shown that many plant phenols (stilbene, curcumin, catechins, flavonoids, etc.) are effective antioxidants that can protect cells during oxidative stress. In addition to direct antioxidant effects, phytophenols can also provide protective effects by activating the Keap1/Nrf2/ARE REDOX-sensitive signaling system and regulating autophagy [58]. The upregulation of phenolic substances in plants after the M treatment group is supposed to enhance the upregulation of phenols in plants to improve the defense ability of plants. Uracil analogues can stimulate the development of plants [59]. This study found that the content of uracil in plants treated with M decreased, which is supposed to be related to the resistance of plants to pathogens. All in all, compared with other treatments, *Trichoderma harzianum* mainly enhances the ability of plants to resist pathogens by stimulating their own defense responses, such as enhancing antioxidant capacity and activating related pathways.

In summary, this study found that the up-regulation or down-regulation of substances in the heat map were all changes that occurred in plants after treatment, and these changes were inevitably related to the resistance of plants to pathogens, which proved that our treatment methods had certain effects. From the above discussion, we can find that the treatment group can induce plants to produce substances useful for resisting pathogens or accumulate beneficial substances thereby improving the disease resistance of *Sorghum sudangrass hybirds* by strengthening the defense ability of the plants themselves.

### 3.4. Amino Acid Metabolism, Secondary Metabolism of Flavonoids and the Jasmonic Acid Pathway May Exert Disease-Resistant Effects

This study showed that the metabolic pathways shared by the four groups, CK vs. ZB, X7 vs. ZB, M vs. ZB, and HX vs. ZB, are the tricarboxylic acid cycle, glutamate–arginine, and tryptophan metabolism pathways (Figure 7). Amino acids have various prominent functions in plants. In addition to their usage during protein biosynthesis, they also represent building blocks for several other biosynthesis pathways and play pivotal roles during signaling processes as well as in plant stress responses [60]. Arginine is biosynthesized in many plants as part of the urea cycle [61]. As urea accumulates significantly during leaf senescence and is utilized for long-range nitrogen transport in the phloem, indicating a comparable function in plants [62], the metabolism of Arginine in plants evidently serves a crucial role in the nitrogen economic cycle. While the content of arginine derivatives was down-regulated in the CK group in the CK vs. ZB group, it showed up-regulation in all three treatment groups (X7, M, HX). In addition, the pathway for the production of pantothenic acid from arginine in the M group treated with *Trichoderma harzianum* distinguished itself from the other groups. Spermine decarboxylase catalyzes the formation of guanidinamine, a precursor in polyamine biosynthesis which includes putrescine, spermidine, and spermine. Previous studies have demonstrated the essential role of polyamines in plant growth, development, and stress response [63]. Therefore, it is hypothesized that in this study, pathways related to spermidine metabolism were activated to combat external stresses caused by plant infestation with pathogens. The decrease in pantothenic acid levels observed after treatment in group M could be attributed to the ability of *T. harzianum* to resist pathogenic fungi independently, without the need to trigger arginine metabolism. Cytoplasmic glutamate plays a crucial role in maintaining the balance of redox reactions and preventing oxidative stress in cells by facilitating the production of glutathione. Additionally, GSH is essential for redox signaling, detoxification of xenobiotics, and regulation of cell processes such as proliferation, apoptosis, immune function, and fibrogenesis [64]. Interestingly, glutamate was down-regulated in all four groups in this study; however, N-phenylacetylglutamate, an important derivative of glutamate, was found to be up-regulated in the pathways of the three treatment groups (X7 vs. ZB, M vs. ZB, and HX vs. ZB). This study speculates that the up-regulation of this precursor substance may facilitate the clearance of excess reactive oxygen species (ROS). In the present study, ascorbic acid was found to be up-regulated in group M. Previous reports [65] indicate that redox changes associated with the ascorbate–glutathione cycle regulate retrograde signaling from chloroplasts and mitochondria to the nucleus, thereby influencing the expression of genes that alter plant growth and defense responses. Additionally, the literature demonstrates that two crucial metabolic pathways, namely the ascorbate–glutathione cycle and the oxidative pentose phosphate pathway, play a role in plant tolerance to biotic stresses following treatment with *Trichoderma harzianum* [66]. Notably, in this study, only group M exhibited the production of the ascorbate–glutathione cycle, which aligns with existing studies on the antioxidant regulation by *Trichoderma harzianum*, suggesting that the presence of the ascorbate–glutathione cycle in *Trichoderma harzianum*-treated plants contributes to their ability to withstand stress.

Tryptophan metabolism generates indole compounds and melatonin, and it also plays a crucial role in inducing signals that mediate various phytochemical and morphogenetic pathways [67]. In the present study, tryptophan derivatives were down-regulated in the CK group within the CK vs. ZB comparison, whereas they exhibited up-regulation in all three other treated groups. Consequently, this study concludes that tryptophan in the treated groups induced plants to generate new pathways in response to stress. Furthermore, tryptophan was analyzed for its role in the production of melatonin and indole compounds, which were produced across all four groups. Indole compounds are significant for their role in plant growth regulation, as they stimulate root and fruit formation while also activating the plant’s immune system against biotic and abiotic stressors [68]. Furthermore, indole compounds exhibited down-regulation in the control (CK) group, whereas the other three treatment groups predominantly displayed up-regulation, with the M group showing complete up-regulation. Consequently, the M group demonstrated superior performance in the tryptophan production pathway for indole synthesis. Melatonin functions as a signaling molecule that mediates the plant defense response against pathogen attacks via the mitogen-activated protein kinase pathway [69]. In this study, the up-regulation of tryptophan in the three treatment groups facilitated melatonin production to enhance defense against pathogen invasion. But, melatonin was down-regulated in the CK group, indicating that all three treatment groups exhibited a protective effect on tryptophan metabolism.

As plant secondary metabolites, flavonoids play essential roles in many biological processes and responses to environmental factors in plants [70]. Studies have shown that phenylpropanoids contribute to plant responses to various biotic and abiotic stresses and are associated with the biosynthesis of lignin and flavonoids [71]. In this study, phenylpropanes were metabolized via the X7 pathway to produce pinocembrin. Pinocembrin is a natural flavonoid compound that can be extracted from honey, propolis, ginger roots, wild marjoram, and various other plants. It has demonstrated the capacity to reduce reactive oxygen species, modulate mitochondrial function, and regulate apoptosis [72]. In the present study, it was hypothesised that group X7 responded to the dilemma of biotic stress by increasing the content of pinocembrin. In the M, HX, and CK groups, naringenin was produced; however, it exhibited down-regulation. Relevant reports [73] indicate that the most significant antioxidant activities of naringenin include scavenging free radicals and preventing lipid peroxidation. Furthermore, studies in animals [74] have demonstrated that naringenin mitigates non-alcoholic fatty liver disease by down-regulating the NLRP3/NF-κB pathway in mice. In conjunction with the present study, it is speculated that naringenin down-regulates to attenuate lipid over-oxidation to ward off stress. In addition, the expression of ferulic acid and cilostazol was up-regulated in both the M and HX groups. According to a study [75], ferulic acid can enhance the activity of key enzymes and gene expression, thereby promoting the synthesis of antioxidants and secondary metabolites, which may improve disease resistance in apples. In the present study, it was hypothesised that both M and HX groups could increase the levels of ferulic acid and cilostazol to reduce the effects of ROS, thereby improving the disease resistance of *Sorghum sudangrass hybrids*.

Lipids are a crucial and diverse class of biomolecules. Their structural heterogeneity in plants is staggering, and many aspects of plant life are manifested and mediated by lipids [76]. Arachidonic acid (AA) is an integral constituent of the biological cell membrane. The four double bonds of AA predispose it to oxygenation, which leads to a plethora of metabolites of considerable importance for the proper function of the immune system as well as the promotion of allergies and inflammation, resolving inflammation [77]. Jasmonates (JAs) have become recognized as one of the main plant hormones that regulate stress responses by activating defense programs and the production of specialized metabolites [78]. Plant resistance to biotrophic pathogens is classically believed to be mediated through salicylic acid (SA) signaling, leading to hypersensitive response followed by the establishment of Systemic Acquired Resistance. In one study, it was suggested that the results of the JA-SA interaction may play an important role in the defence response against the biotrophic pathogens they studied [79]. In our study, we observed that AA was up-regulated in both the M and X7 groups. We speculate that the invasion of pathogens triggers an inflammatory response in the organism, resulting in the up-regulation of AA to mitigate the inflammation. Menawhile, JA-SA was specifically up-regulated in the M group, indicating that their interactions play a crucial role in the acquisition of systemic immunity in the plant, as well as the activation of its defense mechanisms.

The common pathways identified among the four groups—CK vs. ZB, X7 vs. ZB, M vs. ZB, and HX vs. ZB—were associated with glutamate–arginine and tryptophan metabolism. All three treatment groups demonstrated therapeutic effects in these pathways, while the generation of pantothenic acid in the M group set it apart from the other three groups. In the flavonoid metabolic pathway, X7 up-regulated pinocembrin to reduce reactive oxygen species (ROS), whereas naringenin was down-regulated in both the M and HX groups to mitigate lipid over-oxidation. Additionally, ferulic acid and cilostazol were generated to enhance the disease resistance of *Sorghum sudangrass hybrids*. In the lipid metabolism pathway, AA was up-regulated in X7, and JA was up-regulated in HX, while the up-regulation of the JA-SA cycle was regarded as a specific indicator of disease resistance in the M group. Collectively, these findings suggest that group M exerts a more significant influence on the metabolic pathways and plays a crucial role in enhancing disease resistance and improving plant immunity.

In addition, in this experiment, we found that the disease rate of *Sorghum sudangrass hybrids* decreased significantly when the seedlings were first treated with fungicides and then treated again with pathogenicity treatment, so we believe that the addition of the three fungicides is effective in preventing plant diseases. *Trichoderma harzianum* as a biocide has a better fungicide inhibition effect, and in the later stage, we can consider making the biocide into wettable powder for field application to inhibit the occurrence of plant diseases.

## 4. Materials and Methods

### 4.1. Trial Material

In October 2022, the most pathogenic strain of *Colletotrichum boninense* (OR24375) was isolated and purified after observing and sampling leaf spot disease in *Sorghum sudangrass hybrids* grown in Sichuan Province, southwest China (longitude 97°21′ to 108°42′ E, latitude 26°03′ to 34°19′ N). The fungi and bacteria used for the experiments were *Aspergillus albus*-Z7 (MK841443.1), *Trichoderma asperellum*-Z8 (ON209093.1), *Talaromyces funiculosus*-Z9 (MG748629.1), *Trichoderma harzianum*-M (OK445677), and *Talaromyces purpureogenus*-Z19 (MN121629.1), *Bacillus velezensis*-X7 (HQ662594.1), *Bacillus velezensis*-X12 (OQ874287.1), *Bacillus amyloliquefaciens*-X15 (MT579613.1), and *Bacillus subtilis*-X21 (MT233097.1) were obtained from the Grassland Microbiology Laboratory, College of Animal Science and Technology, Southwestern University. A total of 25% pyraclostrobin (Zhongbao Green Farming Group), 70% thiophanate–methyl powder (Zhongbao Green Farming Group), 60% zalfexam powder (Zhongbao Green Agriculture Group), 70% mancozeb powder (Zhongbao Green Agriculture Group), and *Sorghum sudangrass hybrid* seeds (Jiangsu Zhengda Grass Industry) were also used.

### 4.2. Screening of Chemical Fungicides

Four chemical fungicides, pyraclostrobin, zalfexam, thiophanate–methyi, and mancozeb, commonly used to treat leaf spot, were selected and after pre-testing they were sequentially diluted to 20 mg/mL, 10 mg/mL, 5 mg/mL, and 0.1 mg/mL and added to Potato Dextrose Agar (PDA) medium as part of a growth inhibition test. Three replicates were set up for each treatment and sterile H_2_O + PDA medium was used as a control. The pathogenic fungi were formed into 4 mm diameter cakes cultured at 28 °C for 7 days, inoculated in the centre of the mixing and control plates, placed with the fungi surface facing down, marked and then cultured in an inverted incubator at 28 °C, and the colony diameters were measured and recorded using the crisscross method after 3 and 7 days to calculate the minimum inhibitory concentration.

### 4.3. Screening of Biological Fungicides

Fungi were cultured using the plate confrontation method. Pathogenic fungi were cultured at 28 °C for 7 days and were formed into a cake with a 4 mm diameter. They were inoculated in the center of one side of the plate, while Z7, Z8, Z9, Z19, and M were inoculated in the center of the other side. The plates were then inverted in the incubator at 28 °C with the fungal faces facing downwards, labeled, and the diameter of the colonies was measured and recorded after 3 and 7 days. The fungal growth diameter was calculated using the normal growth of the pathogenic fungi as the control, and the inhibition rate was determined by comparing it to the normal growth of the pathogenic fungi.

Bacteria were assessed using the plate delineation confrontation method. The bacteria were cultured at a constant temperature of 28 °C for 7 days, with a cake diameter of 4 mm. They were inoculated in the center of one side of the plate, while the other side was inoculated with X7, X12, X15, and X21, respectively. The diameter of the pathogenic bacteria colonies was measured after culturing at 28 °C for 7 days, and the growth inhibition rate was calculated.

Bacteriostatic inhibition rate = (colony diameter of control group − colony diameter of test group)/colony diameter of control group × 100%

### 4.4. Potting Trials

Preparation of fungal spore solution: The fungus was inoculated on Potato Dextrose Agar (PDA) and incubated at 28 °C for 7 days. The spores were then rinsed with sterile water and the spore concentration was adjusted to 1 × 10^6^ colony-forming units per milliliter (cfu/mL). Subsequently, 1 mL of the spore solution was taken and incubated in 100 mL of Potato Dextrose Broth (PDB) for 48 h. Preparation of bacterial broth: The bacteria were inoculated in liquid Luria–Bertani (LB) medium and incubated for 48 h, resulting in a bacterial concentration of 8 × 10^6^ cfu/mL. Configurations of chemical reagents: A mixture of 1 g of zalfexam agent and 99 mL of water was prepared to obtain a 10 mg/mL solution of zalfexam.

In May 2024, a field trial was conducted at the Rongchang Forage Germplasm Farm of Southwest University in Chongqing (105°17′–105°44′ E, 29°150′–29°41′ N). The area had an average altitude of 308 m, a relative humidity of 72%, and an average annual temperature of 17.8 °C. The soil at the test site was grey–brown–purple soil with a medium loamy texture and a soil pH of 6.5. *Sorghum sudangrass hybrids* were divided into five groups for different treatments. Group ZB received pathogenic treatment by pricking seedlings with sterile needles and watering with pathogenic fungus solution on the first, third, and fifth days in the fourth week. Group M was treated with *Trichoderma harzianum* fungus solution from the roots on the first, third, and fifth days in the third week, followed by the same pathogenic treatment as Group ZB in the fourth week. Group HX was treated with zalfexam agent on the leaves and roots, with spraying on the first, third, and fifth days of the third week. In the fourth week, they received the same pathogenicity treatment as the ZB group. Group X7 was treated with a bacterial solution from the roots on the first, third, and fifth days in the third week, followed by the same pathogenic treatment as Group ZB in the fourth week. The CK group served as a blank control, receiving the same number of waterings with an equal amount of sterile water.

### 4.5. Measurements of Plant Physiological Parameters

Plant height, root length, root volume, root number, leaf length, and other growth indices were measured in each treatment group (CK, ZB, M, HX, and X7) using vernier calipers during the fifth week of growth of *Sorghum sudangrass hybrids*. Each measurement was replicated five times within each group.

### 4.6. Antioxidant Enzyme Activity Assay

In the fifth week, the leaves of *Sorghum sudangrass hybrids* were collected to analyze the activity of plant antioxidant enzymes across various treatment groups. The leaves of *Sorghum sudangrass hybrids* were weighed 1 g from each treatment group and ground into powder. A ten-fold amount (m/V) of phosphate buffer solution (PBS) at pH 7.0 was added to the sample, followed by grinding in an ice bath and centrifugation at 3000 rpm for 15 min. The resulting supernatant was then used for the enzyme activity assay after suitable dilution. Super oxide dismutase (SOD) viability was assessed using the nitrogen blue tetrazolium (NBT) method. One viable unit of SOD was defined as the amount per gram of tissue at 560 nm that corresponded to 50% inhibition in 1 mL of the reaction solution. Peroxidase (POD) activity was quantified using the guaiacol method. One unit of enzyme activity was defined as the increase in absorbance at 470 nm by 0.5 per gram of tissue within the reaction system over a period of 1 min [80]. Catalase (CAT) activity was determined using the UV absorbance method, with one unit of CAT activity defined as the amount of enzyme needed to catalyze the decomposition of 1 μmoL of H_2_O_2_ per gram of tissue per minute at 25 °C. Chitinase activity was determined using the chitinase Assay kit (Solarbio Science & Technology Co., Ltd., Beijing, China), following the instructions from the manufacturer. One unit of enzyme activity (U/g) is defined as the amount of enzyme that breaks down chitin to produce 1 μmoL of N-acetyl-D-(+)-glucosamine per gram of tissue per hour at 37 °C [81].

### 4.7. Sample Preparation of Non-Targeted Metabolome of Sorghum sudangrass hybrids Leaves

Leaf samples from groups ZB, CK, M, X7, and HX were freeze-dried using a freeze-dryer (Songyuan Huaxing Co., Beijing, China) for 24 h and then ground into powder with a high-throughput tissue grinder (Xinzhi Biotechnology Co., Ningbo, China) at 50 Hz. Subsequently, a 50 mg powder sample was mixed with 0.6 mL of 70% methanol and left at 4 °C for 12 h. Following ultrasonic crushing for 5 min (25 KHz–40 KHz), the sample underwent centrifugation at 12,000× *g* for 10 min. Each group of samples had five biological replicates, with 10 μL taken from each group of sample and combined to create a quality control (QC) sample. To ensure method stability and data reliability, one QC sample was inserted for every five samples. LC-MS/MS analyses were carried out using an ultra-high performance liquid chromatography (UHPLC) UltiMate 3000 system (Dionex, Sunnyvale, CA, USA), with instrument parameters and operating procedures derived from previous literature published by the group [3].

### 4.8. Data Analysis

The experimental data on different growth metrics of Gotanda grass were analysed by one-way analysis of variance (ANOVA) using Graphpad Prism 8.0, and the variability of data was expressed as SEM of the mean, and differences were considered significant if *p* < 0.05. Data from untargeted metabolomics were imported into Compound Discoverer 3.2 and after peak detection, extraction, deconvolution, normalisation, and peak alignment, were matched through Mz Cloud and mz Vault databases. The preprocessed data were subjected to multivariate statistical analyses, including PCA and OPLS-DA, on the Mavi Metabolic Platform (https://cloud.metware.cn, accessed on 20 June 2024). Potential differential metabolites were screened according to different projected significance (*VIP* > 1 and *p* < 0.05), and volcano plots and heat maps were produced for statistical analysis of the experiments. Differential metabolite pathways were enriched for important metabolic pathways using Metaboanalyst 5.0 (https://www.metaboanalyst.ca/, accessed on 5 June 2024) to analyse differential metabolite chemicals. In addition, the KEGG database (https://www.kegg.jp/kegg/pathway.html, accessed on 25 June 2024) was used to annotate and construct the pathway.

## 5. Conclusions

To identify an efficient and environmentally friendly method for controlling leaf spot disease in *Sorghum sudangrass hybrids*, this study first screened the most effective biological and chemical fungicides using plate assays. Subsequently, the efficacy of various fungicide treatments on *Sorghum sudangrass hybrids* was evaluated through antioxidant enzyme activity and metabolomic analysis. The following conclusions were drawn:The best chemical inhibitor was zalfexam, with a minimum inhibitory concentration of 10 mg/mL. Among the choices of biocides were the highest inhibition rate of *Bacillus velezensis*, where X7 could reach 51.85%, and the highest inhibition rate of *Trichoderma harzianum,* which was 80.56%.*Trichoderma harzianum*-treated group M showed an increase in both POD and chitinase enzyme activities, zalfexam-treated group HX showed an increase in SOD, POD, and chitinase enzyme activities to defend against the aggression, and the Bacillus velezensis-treated group X7 only showed a significant increase in POD enzymes. This study concludes that all three groups exhibited antioxidant capacities; however, SOD was identified as the primary free radical scavenger, with a significant enhancement in activity observed in group M. Therefore, it can be concluded that group M had a more pronounced influence on antioxidant enzyme properties.The common pathway of the four groups CK vs. ZB, X7 vs. ZB, M vs. ZB, and HX vs. ZB, is glutamate–arginine and tryptophan metabolism, and three of the treatment groups (X7, M, and HX groups) are resistant to stress. The X7 group reduces reactive oxygen species and improves plant disease resistance through tyrosine metabolism, pinocembrin production, and AA up-regulation. The M group responds to ROS and bolster plant resistance via its glutathione–ascorbic acid and tyrosine metabolism, along with the up-regulation of jasmonic acid-salicylic acid (JA-SA) and the production of ferulic acid and cilostazol, which collectively contribute to stress mitigation by eliminating ROS and improving plant disease resistance. In contrast, the HX group primarily increases jasmonic acid content and the production of ferulic acid and cilostazol to withstand stress.This study demonstrates that the combination of bacterial inhibition observed during plate culture, plant growth indices, alterations in antioxidant enzyme activities, and key metabolic pathways indicates that group M, treated with *Trichoderma harzianum*, is more effective for disease control and enhances plant resistance to stress.

This paper presents the first case of utilizing biological and chemical fungicides for the control of leaf spot induced by *Colletotrichum boninense* in *Sorghum sudangrass hybrids*. Ultimately, the environmentally friendly *Trichoderma harzianum* was identified as the most effective fungicide. This study not only addresses the issue of *Colletotrichum boninense* in *Sorghum sudangrass hybrids* but also serves as a valuable reference for managing other crops affected by this pathogen.

## Figures and Tables

**Figure 1 ijms-25-09505-f001:**
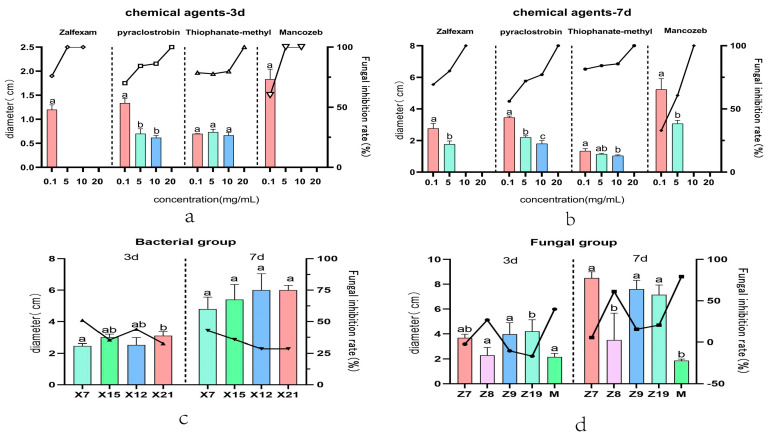
(**a**,**b**) Represent the diameter (histogram) and inhibition rate (line graph) of pathogenic fungal growth after 3 d and 7 d of treatment with four different concentrations of chemicals. (**c**) Represents the diameter (histogram) and inhibition rate (line graph) of four different bacteria after plate confrontation with pathogenic fungi. (**d**) Represents the diameters (histograms) and inhibition rates (line graphs) of five different fungi after confronting the plates with the pathogenic fungal. Note: Same lowercase letters indicate non-significant (*p* > 0.05); different lowercase letters indicate significant differences (*p* < 0.05).

**Figure 2 ijms-25-09505-f002:**
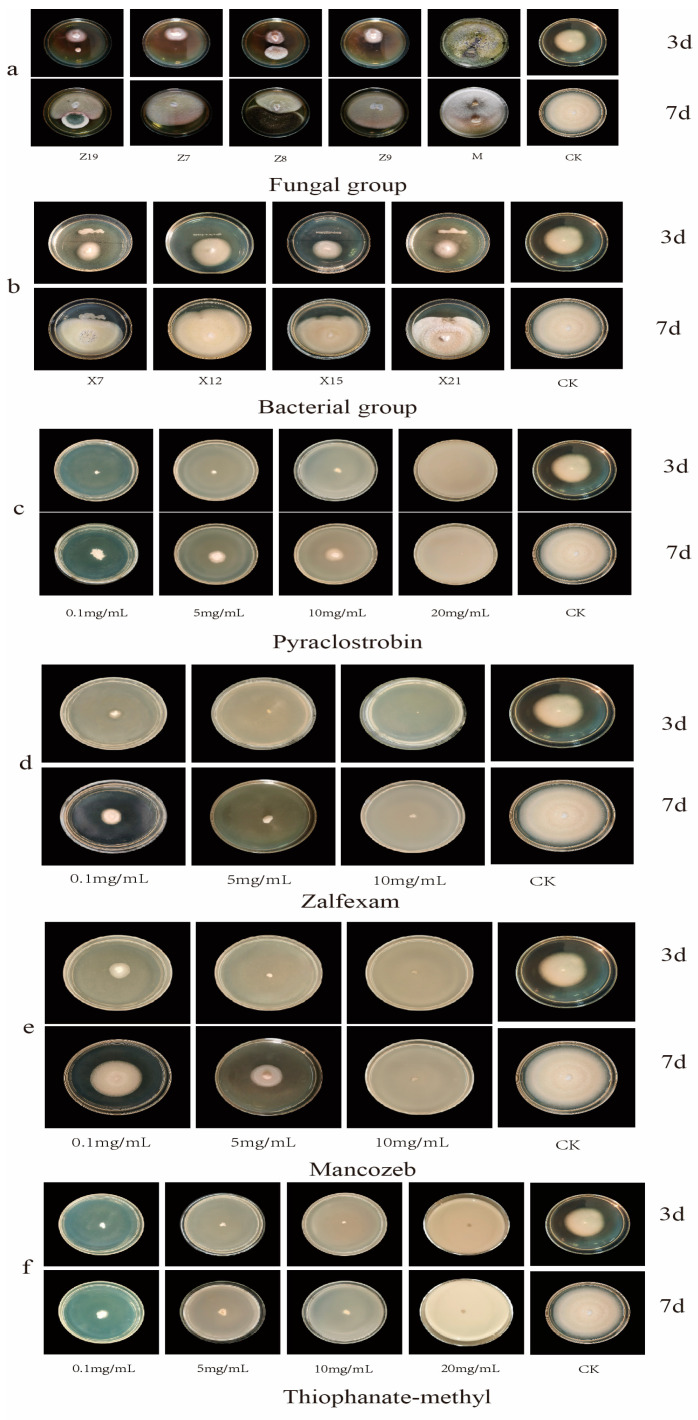
(**a**) Represents the frontal growth of colonies of different fungi and pathogenic fungal 3 d and 7 d after confrontation on PDA. (**b**) Represents the frontal growth of colonies of different bacteria and pathogenic fungal 3 d and 7 d after confrontation on PDA. (**c**–**f**) Represent the colony frontal growth of pathogens on PDA for 3 d and 7 d after addition of different concentrations of chemicals.

**Figure 3 ijms-25-09505-f003:**
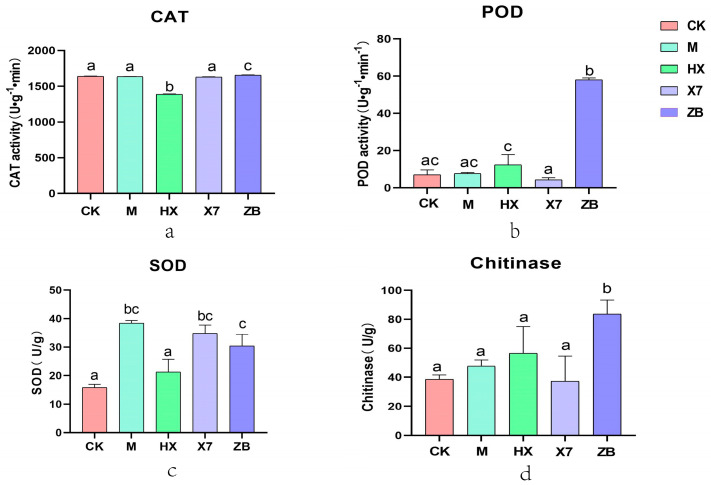
(**a**) Represents five groups in CAT enzyme activity, (**b**) in POD enzyme activity, (**c**) in SOD enzyme activity, and (**d**) in chitinase activity. Note: Same lowercase letters indicate non-significant (*p* > 0.05); different lowercase letters indicate significant differences (*p* < 0.05).

**Figure 4 ijms-25-09505-f004:**
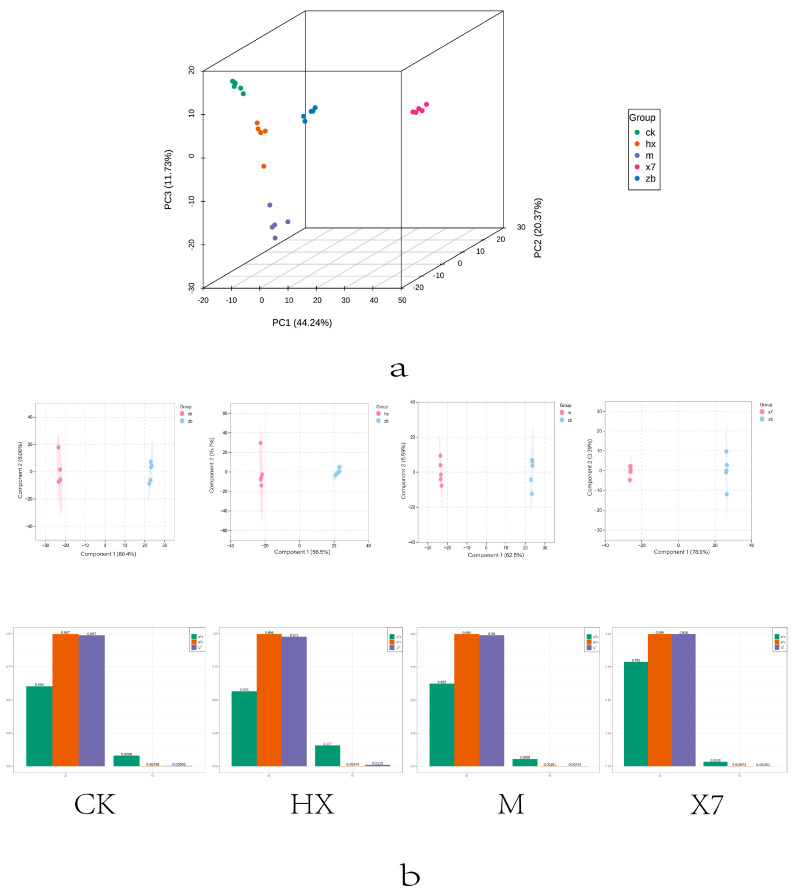
(**a**) Represents PCA plots for the five groups. (**b**) Represents the OPLS−DA score plot versus the model plot.

**Figure 5 ijms-25-09505-f005:**
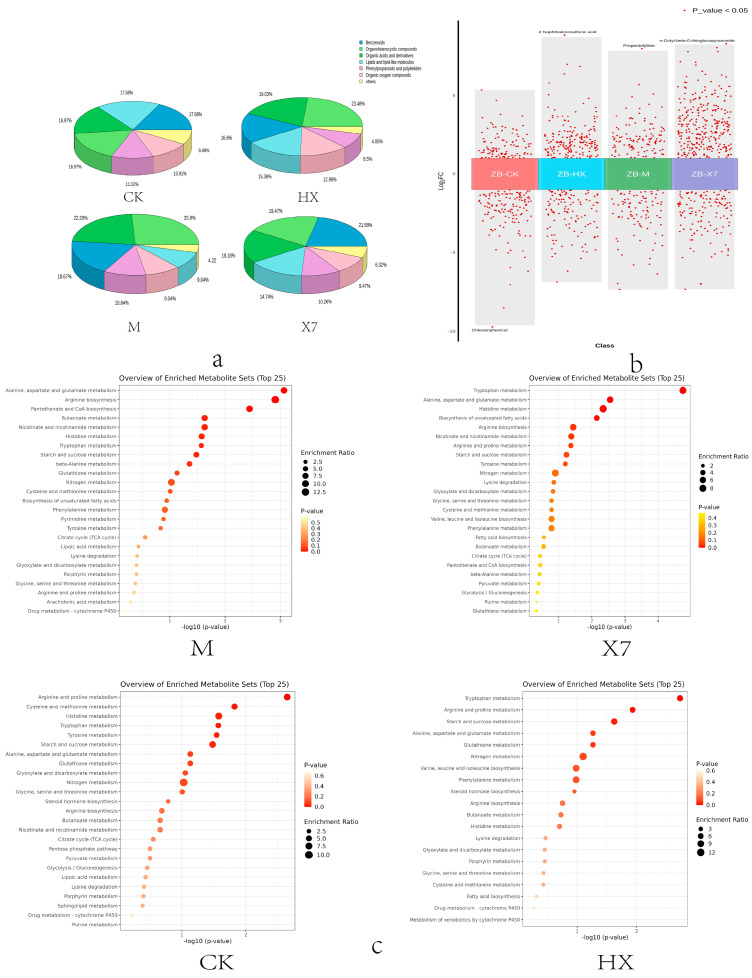
(**a**) Represents 3D pie charts of differential metabolites in the M, X7, CK, and HX groups compared to the ZB group. (**b**) Represents multi-group differential volcano plots. (**c**) Represents metabolite enrichment plots in the M, X7, CK, and HX groups compared to the ZB group.

**Figure 6 ijms-25-09505-f006:**
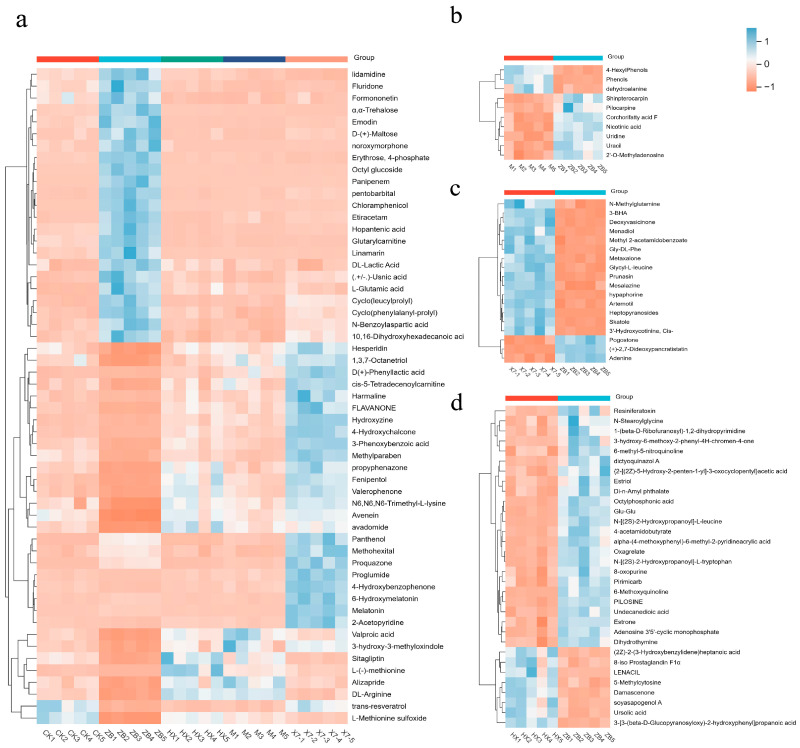
(**a**) Heat map representing differential metabolites common to all five groups. (**b**) Heat map representing differential metabolites specific to group M only. (**c**) Heat map representing differential metabolites unique to group X7 only. (**d**) Heat map representing differential metabolites unique to group HX only.

**Figure 7 ijms-25-09505-f007:**
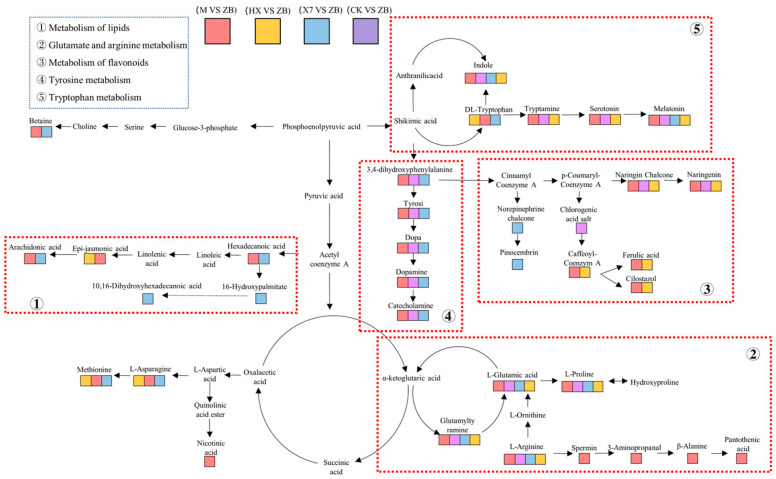
Metabolic pathway.

**Table 1 ijms-25-09505-t001:** Effects of different treatments on the biomass of *Sorghum sudangrass hybrids*.

Groups	Plant Height (cm)	Number of Roots (Pieces)	Root Length (cm)	Root Volume (mL)	Leaf Length (cm)
CK	49.50 ± 2.179 ^a^	6.667 ± 1.528 ^a^	7.333 ± 1.528 ^a^	0.04333 ± 0.005774 ^a^	34.33 ± 3.215 ^a^
ZB	34.33 ± 9.018 ^b^	5.000 ± 1.000 ^a^	6.067 ± 1.290 ^a^	0.01667 ± 0.005774 ^a^	19.00 ± 5.568 ^b^
M	51.83 ± 4.537 ^a^	5.333 ± 1.155 ^a^	7.067 ± 1.888 ^a^	0.06000 ± 0.01732 ^a^	37.00 ± 2.646 ^a^
X7	49.50 ± 2.179 ^a^	7.667 ± 1.155 ^a^	8.333 ± 2.255 ^a^	0.05667 ± 0.02887 ^a^	36.83 ± 3.547 ^a^
HX	50.00 ± 2.179 ^a^	6.000 ± 1.732 ^a^	13.93 ± 6.361 ^a^	0.06000 ± 0.02646 ^a^	35.07 ± 3.614 ^a^

Note: Same lowercase letters indicate non-significant (*p* > 0.05); different lowercase letters indicate significant differences (*p* < 0.05).

## Data Availability

Data are contained within the article.

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
