# Peer review of "Metabolic and Antioxidant Responses of Different Control Methods to the Interaction of Sorghum sudangrass hybrids-Colletotrichum boninense"

_ijms, 2024, doi:10.3390/ijms25179505_

Round 1
Reviewer 1 Report
Comments and Suggestions for Authors
Colletotrichum boninense is the main pathogenic fungus causing leaf spot disease in Sorghum sudangrass hybrids,explored efficient antifungal products is one of the effective means of disease prevention and control. The manuscript identified three effective agents that inhibit Colletotrichum boninense, and the research has certain practical value. However, the research results of the manuscript are incomplete, and further improvement is necessary.
1. The inhibitory effects of Zalfexam, Bacillus velezensis, and Trichoderma harzianum on the pathogenicity of Colletotrichum boninense on the host are not clear, which is the most important point. Inhibiting the growth of pathogenic fungi does not necessarily mean inhibiting their pathogenicity on the host. The author must supplement this experiment, including its preventive and therapeutic effects.
2. The author tested the enzyme activities of CAT, POD, SOD, and Chitinase in plants under different treatments and compared them with the ZB group. What does the author want to clarify? There will definitely be significant differences between pathogen infection and treatment with antifungal agents. Why didn't the author compare the plants treated with antifungal agents and pathogens with the ZB group? Simple antifungal agents treatment can only be compared with the CK group, and comparing with the ZB group is meaningless.
3. Colletotrichum boninense is a pathogenic fungus, in many parts of the manuscript, it is written as pathogenic bacteria, including the vertical axis annotation in Figure 1.
4. What is the concentration of Bacillus velezensis and Trichoderma harzianum used in the article? Has the live bacterial or fungus content been measured before use? Is it used directly after fermentation or diluted? The materials and methods are not clearly stated.
Comments on the Quality of English Language
No comment.
Author Response
Dear Editors and Reviewers:
Thank you for your letter and for the reviewers' comments concerning our manuscript entitled "Metabolic and Antioxidant Responses of Different Control Methods to The Interaction of Sorghum sudangrass hybrids - Colletotrichum boninense" (ID: ijms-3149416). Those comments are all valuable and very helpful for revising and improving our paper, as well as the important guiding significance to our research. We have studied comments carefully and have made correction which we hope to meet with approval.We have marked all changes to the manuscript using red font.The main corrections in the paper and the responds to the reviewer's comments are as flowing:
Responds to the reviewers' comments:
Reviewer 1:
Colletotrichum boninense is the main pathogenic fungus causing leaf spot disease in Sorghum sudangrass hybrids,explored efficient antifungal products is one of the effective means of disease prevention and control. The manuscript identified three effective agents that inhibit Colletotrichum boninense, and the research has certain practical value. However, the research results of the manuscript are incomplete, and further improvement is necessary.
Response 1: First, we very much recognize the suggestions you have made to us.This is a good suggestion to improve the readability of the manuscript. The application of the results of the experiment we will develop in the subsequent experiments.
Comments 1:
- The inhibitory effects of Zalfexam, Bacillus velezensis, and Trichoderma harzianum on the pathogenicity of Colletotrichum boninense on the host are not clear, which is the most important point. Inhibiting the growth of pathogenic fungi does not necessarily mean inhibiting their pathogenicity on the host. The author must supplement this experiment, including its preventive and therapeutic effects.
A: Thank you very much for reviewing our article, in 4.4 potting trials of three fungicides Zalfexam, Bacillus velezensis, and Trichoderma harzianum on plants. We treated Sorghum sudangrass hybrids seedlings with Zalfexam, Bacillus velezensis, and Trichoderma harzianum in the third week followed by pathogenicity treatment in the fourth week, and the incidence of the disease after 7 days of inoculation was 12%, 22.8%, and 24% per cent in groups M, HX, and X7, and 67.8% in group ZB.
2.The author tested the enzyme activities of CAT, POD, SOD, and Chitinase in plants under different treatments and compared them with the ZB group. What does the author want to clarify? There will definitely be significant differences between pathogen infection and treatment with antifungal agents. Why didn't the author compare the plants treated with antifungal agents and pathogens with the ZB group? Simple antifungal agents treatment can only be compared with the CK group, and comparing with the ZB group is meaningless.
A: Thank you very much for your suggestion, your idea coincides with ours, in the 4.4Potting trials in this article we treated different plants with fungicides followed by pathogenicity treatments, and then compared the three treatment groups with the ZB group, as this article did not express clearly to you to a certain degree of distress, so we are again in the article to modify and add.
- Colletotrichum boninense is a pathogenic fungus, in many parts of the manuscript, it is written as pathogenic bacteria, including the vertical axis annotation in Figure 1.
A: Thank you very much for raising this issue, we have uniformly replaced pathogenic bacteria with pathogenic fungus in our articles, at the same time we also modify Figure 1and have double-checked and double-checked the details of that article.
4.What is the concentration of Bacillus velezensis and Trichoderma harzianum used in the article? Has the live bacterial or fungus content been measured before use? Is it used directly after fermentation or diluted? The materials and methods are not clearly stated.
A:Thank you very much for pointing this out, the concentration of bacterial fluid we used in the article is mentioned in 4.4Potting trials, preparation of fungal spore solution: the fungus was inoculated on Potato Dextrose Agar (PDA) and incubated at 28℃ for 7 days. The spores were then rinsed with sterile water and the spore concentration was adjusted to 1×106 colony-forming units per milliliter (cfu/mL). Subsequently, 1 mL of the spore solution was taken and incubated in 100 mL of Potato Dextrose Broth (PDB) for 48 hours. Preparation of bacterial broth: The bacteria were inoculated in liquid Luria-Bertani (LB) medium and incubated for 48 h, resulting in a bacterial concentration of 8×106 cfu/mL. I'm sorry it wasn't detailed in the article, we've refined that section in the article. Thanks again for your advice and have a happy, safe and joyful life.
Reviewer 2 Report
Comments and Suggestions for Authors
Minor editing of English language required.
1. The paper is well-organized and presents a clear, logical flow from the introduction to the
conclusion.
2. There are some typos mistakes in the paper in different places. e.g. in line number 9 after
colon there should be a space. Same for line number 100 should be space after full stop.
Table 1 has more spaces in the value. Please revise the manuscript and correct these all
grammatical and typos mistakes.
3. The methodology section could include more details about the statistical analyses used to
interpret the data.
4. It would be beneficial to discuss how the findings might be translated into breeding
programs or field applications.
5. The literature review in the discussion and introduction section could be more focused on
recent developments and literature to better frame the study's contribution.
6. Please explain figure 6 more. Explain more about what inside labeling is.
7. It would be good if proceeded with the field work to validate the findings.
Minor editing of English language required.
Author Response
Dear Editors and Reviewers:
Thank you for your letter and for the reviewers' comments concerning our manuscript entitled "Metabolic and Antioxidant Responses of Different Control Methods to The Interaction of Sorghum sudangrass hybrids - Colletotrichum boninense" (ID: ijms-3149416). Those comments are all valuable and very helpful for revising and improving our paper, as well as the important guiding significance to our research. We have studied comments carefully and have made correction which we hope to meet with approval.We have marked all changes to the manuscript using red font.The main corrections in the paper and the responds to the reviewer's comments are as flowing:
Reviewer 2:
Minor editing of English language required.
Response 2: We would like to express our heartfelt thanks to you for reviewing our article in your busy schedule, as well as for recognising the results of our work, and we wish you all the best in your work and life.
Comments 2:
1.The paper is well-organized and presents a clear, logical flow from the introduction to the conclusion.
A: Thank you for your recognition of our work, it makes us feel very happy and we hope you can be all right and have a good life.
2.There are some typos mistakes in the paper in different places. e.g. in line number 9 after colon there should be a space. Same for line number 100 should be space after full stop.Table 1 has more spaces in the value. Please revise the manuscript and correct these all grammatical and typos mistakes.
A: The details you have raised are very important to us and we have already changed the parts of the article you have mentioned and reviewed and revised the content of the article. Thank you again for your comments.
3.The methodology section could include more details about the statistical analyses used to interpret the data.
A:Thank you very much for your suggestion, it has been revised in the data analysis section of this paper 4.8. data analysis.
4.It would be beneficial to discuss how the findings might be translated into breeding programs or field applications.
A:Your suggestion is very good, and it is the focus of our work after that, and I will add to that part of the discussion.
5.The literature review in the discussion and introduction section could be more focused on recent developments and literature to better frame the study's contribution.
A:Thank you very much for your suggestion, we have added recent references.
6. Please explain figure 6 more. Explain more about what inside labeling is.
A: Thank you very much for pointing this out, we have changed both the image and the legend in this article.
7.It would be good if proceeded with the field work to validate the findings.
A: In fact, this coincides with our idea. In the manuscript we have found a reduction in plant pathogenicity through field trials with Zalfexam, Bacillus velezensis, and Trichoderma harzianum, in addition to the use of Bacillus in the group's fava beans, which also reduces plant pathogenicity, and we will be carrying out more practices to improve plant yields in the future.
Reviewer 3 Report
Comments and Suggestions for Authors
A well written manuscript, minor comments only
line 610 ' Colletotrichum boninense (OR24375) was isolated and purified' - how was this done?
line 625 'pyraclostrobin, zalfexam, thiophanate-methyi and mancozeb, commonly used to treat leaf spot' - what are the recommended field application rates for the purpose of comparison with the laboratory applications?
Author Response
Dear Editors and Reviewers:
Thank you for your letter and for the reviewers' comments concerning our manuscript entitled "Metabolic and Antioxidant Responses of Different Control Methods to The Interaction of Sorghum sudangrass hybrids - Colletotrichum boninense" (ID: ijms-3149416). Those comments are all valuable and very helpful for revising and improving our paper, as well as the important guiding significance to our research. We have studied comments carefully and have made correction which we hope to meet with approval.We have marked all changes to the manuscript using red font.The main corrections in the paper and the responds to the reviewer's comments are as flowing:
Review 3 :
A well written manuscript, minor comments only.
Response 3: Thank you for your recognition of our work. We would like to express our sincere gratitude for reviewing our articles during your busy schedule, really appreciate your review. I wish you success in your work, peace and joy.
1.line 610 ' Colletotrichum boninense (OR24375) was isolated and purified' - how was this done?
A: Thank you for pointing this out.The laboratory pre-studied the leaf spot of Sorghum sudangrass hybrids in Bazhong area and found that the main pathogens were Colletotrichum boninense, Nigrospora sphaerica and Didymella corylicola, among which C. boninense had the strongest pathogenicity ability[1]. Based on published laboratory articles, the specific test steps are as follows: infected tissues (3 mm × 3 mm) at the junction of disease and health of the leaf blades were cut, sterilized in 75% ethanol for 30 s, rinsed in sterile water, and then immersed with 1% sodium hypochlorite for 3 min, followed by rinsing and swabbing. The segments were then placed on PDA at 25 °C. After 3 d of incubation, mycelium growing at the edge was transferred to fresh PDA and cultured at 25 °C. All purified isolates were stored on PDA slant at 4 °C.
2.line 625 'pyraclostrobin, zalfexam, thiophanate-methyi and mancozeb, commonly used to treat leaf spot' - what are the recommended field application rates for the purpose of comparison with the laboratory applications?
A:Thank you very much for your question. Field use of pyraclostrobin is 450-600/ha, the field dosage of zalfexam is 60-100 g of 60% water dispersible granules per acre, the field dosage of thiophanate-methyi is 100-200 g/mu,the field dosage of mancozeb is 176-226 g/mu.
Round 2
Reviewer 2 Report
Comments and Suggestions for Authors
All good. Accepted for publication. best wishes
Author Response
Dear Reviewer:
Thank you very much for your wishes, I hope you can be happy and have a good life!
Reviewer 3 Report
Comments and Suggestions for Authors
Comments addressed
Author Response

(The authors gave the same response as above.)
